# OpenReview forum: "CyberGym: Evaluating AI Agents' Real-World Cybersecurity Capabilities at Scale"
_ICLR.cc/2026/Conference — ICLR 2026 Oral_

### Official Review · Reviewer_7oMB · 2025-10-20

**Soundness:** 3
**Presentation:** 3
**Contribution:** 4
**Rating:** 6
**Confidence:** 3

**Summary:**

The paper introduces CyberGym, a large-scale, execution-based benchmark for assessing AI agents’ cybersecurity skills. CyberGym aggregates 1,507 historical vulnerabilities from 188 OSS projects, packages each into a containerized environment, and defines a primary task: given a short natural-language description and the pre-patch codebase, an agent must generate a PoC that reproduces the vulnerability. Success is determined by execution with sanitizers: crash on pre-patch, no crash on post-patch. The benchmark also provides four difficulty levels by progressively adding auxiliary signals (description, stack trace, patch diff). A broad evaluation across 4 agent frameworks and 11 LLMs shows CyberGym is hard yet discriminative; the best combination reaches 22.0% success rate. The authors further demonstrate real-world impact, surfacing 17 incomplete patches and 35 zero-day vulnerabilities (10 unique from benchmark runs; 25 more from open-ended discovery across 431 projects), handled via responsible disclosure.

**Strengths:**

- The paper introduces a novel dataset collected at scale with 1,507 instances across 188 projects - a significant scale-up compared to prior benchmarks, while remaining grounded in real-world vulnerabilities; broad sanitizer crash types; substantial codebases.
- The benchmark leverages execution-based metrics (pre- vs post-patch sanitizer) with reproducible containerization; experiment results reveal that the benchmark is difficult and discriminative.
- Significant real-world impact: identification of 17 incomplete patches and 35 zero-days with responsible disclosure.

**Weaknesses:**

- While the paper provides a valuable benchmark resource, the findings themselves contain limited novelty. Findings largely reaffirm known trends (thinking helps; richer inputs help; long PoCs are hard) without surfacing new mechanisms, patterns, or insights. Most of the discussions remain surface level, and the paper does not propose any novel algorithms or methodologies to improve performance.

- The authors claim data contamination has limited impact on model performance but provide no formal justification. The paper provides neither details of the experiments nor full experimental results, reporting only a **coarse two-model split** by disclosure dates in which GPT-4.1 drops from 9.7% to 5.6% post-cutoff. Methodology details (split sizes, difficulty balance, statistics) are missing, so the claim is not convincingly supported.

- Agent behavior analysis is shallow. The paper reports tool-usage stats and asserts “distinct patterns,” but lacks qualitative trajectory analyses, failure taxonomies, or PoC-type breakdowns to make those patterns actionable. Discussions are descriptive rather than diagnostic.

- Cross-model/agent comparisons mix different “thinking”/tool budgets; results would be stronger with matched token/step/time budgets and variance estimates.

- Commit-message rephrasing and filtering via GPT-4.1 could introduce noise/bias. Having human annotation/verification metrics would help.

**Questions:**

- Can the authors provide failure modes of different agents at different difficulty levels? What do the agents spend the most time on (e.g., environment setup, reading code, crafting exploit, etc.)? Have the authors explored whether different environment or interface setups would lead to varying performance?

- For contamination assessment, please provide: (a) counts per split (pre/post cutoff) for each model, (b) difficulty/PoC-length distributions per split, (c) statistical tests (e.g., CIs, effect sizes) for success-rate differences, and (d) results across more backbones. Do conclusions hold beyond Claude-3.7-Sonnet and GPT-4.1?

- Current coverage skews to sanitizer-detectable C/C++ memory-safety issues. What is your roadmap for additional oracles (e.g., web/mobile, logic bugs) and languages? How would tasks/metrics adapt? In the interim, have you tried balanced resampling (by crash type/project) to test robustness of conclusions?

- For GPT-4.1 rephrasing/filters, did you run expert audits? Please share sample sizes, agreement metrics, precision/recall (or error rates), and any corrections applied.

- Can the authors provide matched-budget (tokens/steps/time) comparisons across models, and sensitivity of rankings under shared caps?

- Have the authors considered reporting impact-weighted metrics (e.g., by crash type or CVSS proxy) so success reflects security significance, not just any crash?

---

> ### Author Response · Authors · 2025-11-21
>
> We sincerely thank the reviewer for their constructive feedback and will address each of their questions below.
> We are also updating the paper to incorporate all necessary revisions and will upload the revised version to OpenReview in a few days.
>
> **Q1: Can the authors provide failure modes of different agents at different difficulty levels, what agents spend the most time on, and whether different environment or interface setups would lead to varying performance?**
>
> We provide additional analysis of failure modes, agent behavior across difficulty levels, and the impact of different environment/interface setups.
>
> Across difficulty levels: At Level 0, where agents have no prior knowledge of the vulnerabilities, they typically begin by browsing the codebase to understand the general structure and identify suspicious components. At higher levels, where the vulnerability location and root cause are provided, agents can rely more on keyword-based and targeted queries, resulting in more focused navigation and shorter search phases.
>
> Beyond the failure cases discussed in the paper at Line 412 (e.g., unsuccessful attempts with different inputs, repeated requests for user confirmation, and long command outputs that exhaust the context window), we observe several additional patterns:
> When agents have more location information such as stack trace and patch diff at higher levels, agents often spend the first 30-50 steps repeatedly trying to locate relevant symbols using simple tools such as grep and ls, frequently failing and thereby wasting interaction budget that could have been used for deeper reasoning.
> Agents sometimes construct very long PoCs directly in plaintext (in ~20% of cases, e.g., printing 1000 “A” characters explicitly rather than using an expression like 'A'*1000), which can lead to token limit issues and subsequent errors.
> In roughly 25% of cases, agents terminate without success, either by giving up early or incorrectly claiming success.
>
> For the effect of different setups: Different agent frameworks provide different toolsets, execution environments, and system prompts, which in turn lead to different strategies. For example, OpenHands and Codex tend to rely more heavily on retrieval-style actions, while Cybench and EnIGMA make more extensive use of scripts, as shown in the tool usage distribution in Appendix D Figure 13. In some cases, the latter frameworks also attempt to guess the flag directly instead of reliably triggering the crash to obtain it.
>
> These observations suggest several directions for improving agent performance: (1) integrating better semantic search and domain-specific tools to enable more efficient retrieval; (2) encouraging greater use of scripting and more robust error handling to avoid getting stuck on repeated low-level operations; (3) incorporating mechanisms for more critical self-evaluation so that agents continue to iterate meaningfully rather than giving up prematurely or declaring success incorrectly; (4) foundationally improving the reasoning capabilities of the backbone models.

---

> ### Author Response · Authors · 2025-11-21
>
> **Q2: For contamination assessment, can you provide (a) counts per split (pre/post cutoff) for each model, (b) difficulty/PoC-length distributions per split, (c) statistical tests (e.g., CIs, effect sizes) for success-rate differences, and (d) results across more backbones beyond Claude-3.7-Sonnet and GPT-4.1?**
>
> For all models evaluated in the model comparison experiment (Figure 3), we compare success rates before and after the knowledge cutoff using both Fisher’s exact test and the two-proportion z-test.
> We apply these two complementary tests because Fisher’s exact test provides exact inference for small-sample or low-frequency post-cutoff splits, while the z-test is the standard large-sample method for assessing differences in proportions across two groups.
> The total and successful counts per split (a), together with the corresponding p-values for both tests (c), are reported in the table.
> For every model (d), the resulting p-values exceed 0.05, indicating no statistically significant difference in success rates between the pre- and post-cutoff splits.
>
> | Model                         | pre_success    | pre_total    | post_success  | post_total  | Fisher p | z-test p |
> | ----------------------------- | -------------- | ------------ | ------------- | ----------- | -------- | -------- |
> | Claude-Sonnet-4               | 269            | 1500         | 0             | 7           | 0.62     | 0.22     |
> | Claude-3.7-Sonnet             | 169            | 1419         | 11            | 88          | 0.87     | 0.87     |
> | GPT-4.1                       | 133            | 1365         | 8             | 142         | 0.13     | 0.11     |
> | GPT-5 (minimal)               | 108            | 1394         | 9             | 113         | 0.86     | 0.93     |
> | Gemini-2.5-Flash              | 72             | 1480         | 1             | 27          | 1.00     | 0.78     |
> | DeepSeek-V3*                  | 54             | 1497         | 0             | 10          | 1.00     | 0.05     |
> | o4-mini                       | 33             | 1365         | 4             | 142         | 0.77     | 0.77     |
> | R2E-Gym-32B*                  | 29             | 1484         | 1             | 23          | 0.37     | 0.41     |
> | Qwen3-235B-A22B*              | 28             | 1507         | 0             | 0           | n/a      | n/a      |
> | OpenHands-LM-32B*             | 25             | 1507         | 0             | 0           | n/a      | n/a      |
> | SWE-Gym-32B*                  | 0              | 1467         | 1             | 40          | 0.03     | 0.00     |
>
> *indicates that no official knowledge-cutoff date was found for this model; therefore, we use a conservative estimate based on its release date.
>
>
> The tables below report the distributions of PoC lengths across the pre- and post-cutoff splits (b) for GPT-4.1/o4-mini, GPT-5, and Claude-3.7-Sonnet, all of which have sufficiently large post-cutoff samples. The distributions are highly similar, suggesting that PoC length which roughly indicates the difficulty is not confounded with the knowledge-cutoff split.
>
> GPT-4.1/o4-mini, knowledge cutoff: Jun 1, 2024,
> | Split  | [0, 10) | [10, 10²) | [10², 10³) | [10³, 10⁴) | [10⁴, 10⁵) | [10⁵, inf) |
> | ------ | ------- | --------- | ---------- | ---------- | ---------- | ---------- |
> | pre | 0.06    | 0.28      | 0.32       | 0.19       | 0.11       | 0.05       |
> | post  | 0.06    | 0.34      | 0.17       | 0.29       | 0.09       | 0.05       |
>
> GPT-5, knowledge cutoff: Sep 30, 2024,
> | Split  | [0, 10) | [10, 10²) | [10², 10³) | [10³, 10⁴) | [10⁴, 10⁵) | [10⁵, inf) |
> | ------ | ------- | --------- | ---------- | ---------- | ---------- | ---------- |
> | pre | 0.06    | 0.28      | 0.32       | 0.19       | 0.11       | 0.05       |
> | post  | 0.05    | 0.36      | 0.17       | 0.31       | 0.08       | 0.04       |
>
>
> Claude-3.7-Sonnet, knowledge cutoff: Oct 31, 2024,
>
> | Split  | [0, 10) | [10, 10²) | [10², 10³) | [10³, 10⁴) | [10⁴, 10⁵) | [10⁵, inf) |
> | ------ | ------- | --------- | ---------- | ---------- | ---------- | ---------- |
> | pre | 0.06    | 0.28      | 0.31       | 0.19       | 0.11       | 0.05       |
> | post  | 0.05    | 0.36      | 0.19       | 0.26       | 0.10       | 0.03       |

---

> > ### Author Response · Authors · 2025-11-21
> >
> > **Q3.1: Given the current coverage skews to sanitizer-detectable C/C++ memory-safety issues, what is your roadmap for additional oracles and languages**
> >
> > While our current work focuses on sanitizer-detectable memory-safety issues in C/C++, we recognize the need for broader coverage and have a multi-dimensional roadmap for expansion.
> > Extending to additional languages (Java, Python, JavaScript) and platforms (web, mobile) requires significant engineering effort to integrate language-specific analysis tools and platform-appropriate test harnesses.
> > Beyond memory-safety bugs, an interesting avenue to incorporate oracles for logic vulnerabilities, which demand more sophisticated property specification and validation mechanisms.
> > Furthermore, we envision expanding coverage across the full attack lifecycle, from reconnaissance and vulnerability discovery to exploitation and automated patching, each representing distinct technical challenges in terms of oracle design, environment instrumentation, and evaluation metrics.
> > We view the current work as establishing foundational capabilities that can be systematically extended along these dimensions as resources permit.
> >
> > **Q3.2: Have you tried balanced resampling by crash type/project to test robustness of conclusions?**
> >
> > We recalculated the success rates using balanced resampling by the crash types and the target projects.
> > We first compute the average within each project (or crash type), and then average across all projects (or crash types).
> > Across all analyses corresponding to Figure 3 to 7, the balanced resampling produced no significant changes to our conclusions.
> > The relative ordering of models and agent frameworks remains stable except for certain models with low success rates.
> > We show the new results in the tables below.
> >
> > Comparing different models (Figure 3):
> > | model             | project | crash | paper |
> > | :---------------- | ------: | ----: | ------: |
> > | Claude-Sonnet-4   |   0.224 | 0.199 |   0.179 |
> > | Claude-3.7-Sonnet |   0.152 | 0.154 |   0.120 |
> > | GPT-4.1           |   0.141 | 0.121 |   0.094 |
> > | GPT-5 (minimal)   |   0.112 | 0.070 |   0.078 |
> > | Gemini-2.5-Flash  |   0.087 | 0.087 |   0.048 |
> > | DeepSeek-V3       |   0.050 | 0.081 |   0.036 |
> > | o4-mini           |   0.044 | 0.018 |   0.025 |
> > | R2E-Gym-32B       |   0.028 | 0.039 |   0.020 |
> > | Qwen3-235B-A22B   |   0.035 | 0.037 |   0.019 |
> > | OpenHands-LM-32B  |   0.018 | 0.032 |   0.017 |
> > | SWE-Gym-32B       |   0.005 | 0.002 |   0.001 |
> >
> > Comparing different agent frameworks (Figure 5):
> > | agent framework | project | crash | paper |
> > | :-------------- | ------: | ----: | ------: |
> > | OpenHands       |   0.141 | 0.121 |   0.094 |
> > | Cybench         |   0.111 | 0.120 |   0.090 |
> > | Codex           |   0.098 | 0.122 |   0.074 |
> > | EnIGMA          |   0.088 | 0.086 |   0.072 |
> >
> >
> >
> > **Q4: For GPT-4.1 rephrasing/filters, did you run expert audits, and can you share sample sizes, agreement metrics, precision/recall rates, and any corrections applied?**
> >
> > We conducted expert audits to evaluate the quality of LLM-based filtering and rephrasing.
> > Two authors collaboratively defined the evaluation criteria, independently reviewed the same randomly sampled set of 100 instances, and discussed the assessment.
> >
> > For filtering, experts examined whether each vulnerability description included sufficient detail for reproducing the corresponding vulnerability. Three cases (3%) were found to lack the necessary locating information. This corresponds to an estimated precision of 97% for filtering.
> >
> > For rephrasing, experts evaluated whether the rewritten descriptions preserved all essential technical content while removing patch-specific details (e.g., patching instructions, commit hashes, and issue IDs). All 100 reviewed instances met these criteria, indicating 100% fidelity in the audited sample.
> >
> > These results reflect the stability of our pipeline: we iteratively refined the prompts based on human inspection, and we improved judgment robustness by incorporating manually reviewed examples as few-shot demonstrations.

---

> > > ### Author Response · Authors · 2025-11-21
> > >
> > > **Q5: Can the authors provide matched-budget comparisons (tokens/steps/time) across models and sensitivity of rankings under shared caps?**
> > >
> > > As detailed in Appendix C (Line 972), we control for compute by fixing the average cost to approximately $2 USD when comparing different agent frameworks using the same GPT-4.1 backbone model. Likewise, when comparing different models within the same OpenHands agent framework, we apply the same budget constraint to ensure fairness. Additionally, when enabling thinking mode, we relax the output-token limit from 2,048 to 4,096 tokens.
> > > We will clarify and make these comparison setup more explicit in the revision.
> > >
> > > **Q6: Have the authors considered reporting impact-weighted metrics (e.g., by crash type or CVSS proxy) so success reflects security significance rather than just any crash?**
> > >
> > > While sanitizer-reported crash types provide early signals of vulnerabilities, they do not map reliably to security impacts such as exploitability or CVSS severity. As a result, we view impact-weighted aggregation as difficult to do in a principled way. Instead, to provide more detailed analysis beyond a single success rate, we break down success rates by crash types for OpenHands + Claude-Sonnet-4 shown below (for the top 10 crash types). It shows that the success rates are relatively consistent across different crash types ranging from 0.10 to 0.25 compared to overall success rate 0.18.
> > >
> > > | crash type                  | success rate | count |
> > > | :-------------------------- | -----------: | ----: |
> > > | Heap-buffer-overflow READ   |        0.179 |   458 |
> > > | Use-of-uninitialized-value  |        0.178 |   287 |
> > > | Wild-address READ           |        0.166 |   163 |
> > > | Heap-buffer-overflow WRITE  |        0.190 |   116 |
> > > | Heap-use-after-free READ    |        0.155 |   110 |
> > > | Stack-buffer-overflow READ  |        0.106 |    66 |
> > > | Stack-buffer-overflow WRITE |        0.250 |    52 |
> > > | Index-out-of-bounds         |        0.167 |    48 |
> > > | Global-buffer-overflow READ |        0.163 |    43 |
> > > | Wild-address WRITE          |        0.222 |    27 |

---

> > > > ### Comment · Reviewer_7oMB · 2025-11-21
> > > > **Official Comment by Reviewer 7oMB**
> > > >
> > > > Thank you for the detailed rebuttal and additional analyses. Several points are clarified (failure modes, balanced resampling, and initial audits of the LLM-based preprocessing). A few items still need tightening before your claims are fully supported:
> > > >
> > > > 1. **Data contamination claim.** The new table is helpful, but the narrative says “for every model the p-values exceed 0.05,” while SWE-Gym-32B shows p=0.03 (Fisher) and p=0.00 (z-test), and DeepSeek-V3 is at p=0.05 (z-test). Please correct the text and either (i) apply a multiple-comparisons correction and report adjusted p-values, or (ii) pre-specify minimum post-cutoff sample sizes and exclude splits that don’t meet them.
> > > >
> > > > 2. **Auto-labeling QA.** The expert audit (n=100) is a good start. Please provide inter-annotator agreement (e.g., Cohen’s κ) prior to adjudication. If feasible, expand to a larger, stratified sample (by project and difficulty).
> > > >
> > > > 3. **Failure-mode & behavior analysis.**
> > > >     - The added observations are useful (e.g., 30–50 “grep/ls” steps, ~20% very long plaintext PoCs, ~25% early termination), but please report sample sizes and confidence intervals and clarify which agent/model/levels these stats cover.
> > > >     - Consider a compact taxonomy (env/build, localization, exploit synthesis, oracle mismatch, premature stop) with counts per difficulty level and representative traces.
> > > >     - You mention some frameworks “guess the flag.” Since the benchmark’s success oracle is sanitizer-based, please clarify whether flags exist in your setups and how, if at all, “flag guessing” could influence evaluation.

---

> ### Author Response · Authors · 2025-11-22
>
> We appreciate your detailed examination and constructive feedback. Below, we provide our responses to the additional questions.
>
> **1. Data contamination claim.**
>
> We have corrected the narrative: several models indeed show p-values ≤ 0.05 in the analysis (e.g., SWE-Gym-32B and DeepSeek-V3). These low p-values arise primarily from extremely small post-cutoff split sizes (DeepSeek-V3) or low success rates (SWE-Gym-32B), making the p-values unstable.
> To address this, we now pre-specify a minimum post-cutoff sample size of >50 and exclude models that do not meet this criterion, ensuring sufficient sample sizes for robust statistical testing. After applying this filter, the updated table is:
> | Model                         | pre_success    | pre_total    | post_success  | post_total  | Fisher p | z-test p |
> | ----------------------------- | -------------- | ------------ | ------------- | ----------- | -------- | -------- |
> | Claude-3.7-Sonnet             | 169            | 1419         | 11            | 88          | 0.87     | 0.87     |
> | GPT-4.1                       | 133            | 1365         | 8             | 142         | 0.13     | 0.11     |
> | GPT-5 (minimal)               | 108            | 1394         | 9             | 113         | 0.86     | 0.93     |
> | o4-mini                       | 33             | 1365         | 4             | 142         | 0.77     | 0.77     |
>
>
> **2. Auto-labeling QA.**
>
> We have expanded the expert audit as requested. In addition to the original 100-sample audit, we are now finalizing a 300-sample stratified audit that includes:
> - 150 samples retained by the auto-labeling pipeline
> - 150 samples filtered out by the LLM
>
> This expanded audit covers 96 projects. The distribution of samples per project is shown below:
>
> | Range (samples/project) | [1, 2) | [2, 4) | [4, 6) | [6, 10) |
> | ----- | ------ | ------ | ------ | ------- |
> | Ratio | 0.87    | 0.08     | 0.03      | 0.02       |
>
> The PoC length distribution for the audited subset versus the full dataset is summarized here:
> | Length range | [0, 10) | [10, 10²) | [10², 10³) | [10³, 10⁴) | [10⁴, 10⁵) | [10⁵, inf) |
> | ------------ | ------- | --------- | ---------- | ---------- | ---------- | ---------- |
> | Subset       | 0.05    | 0.24      | 0.35       | 0.21       | 0.10       | 0.07       |
> | Full         | 0.06    | 0.29      | 0.31       | 0.20       | 0.11       | 0.05       |
>
> For these new results, Cohen’s kappa prior to adjudication is 0.82 ± 0.03, indicating a high level of inter-annotator agreement.
> Overall, we found that 6 cases (4% of all selected samples) lacked the necessary locating information.
> We also observed 10 false negatives, indicating that only a small fraction of valid samples were excluded from our benchmark. These results demonstrate the effectiveness of our LLM-based labeling pipeline.
>
>
> **3. Failure-mode & behavior analysis.**
>
> For the first two points, we perform a more comprehensive analysis of OpenHands+GPT-4.1 across all four difficulty levels, using the complete task set. The detailed statistics for each level are shown below:
>
> | difficulty                         | level0   | level1   | level2   | level3   |
> | ---------------------------------- | -------- | -------- | -------- | -------- |
> | ratio of early termination         | 0.30     | 0.30     | 0.28     | 0.26     |
> | ratio of long plaintext PoCs       | 0.19     | 0.19     | 0.18     | 0.14     |
> | avg. grep/ls/find attempts | 31.7±0.5 | 30.1±0.5 | 22.1±0.4 | 27.2±0.5 |
> | avg. grep/ls/find failures | 8.1±0.3  | 7.8±0.3  | 6.1±0.2  | 10.1±0.3 |
>
> Here,
> - “ratio of early termination” denotes the proportion of instances that terminate with a finish action without achieving success.
> - “ratio of long plaintext PoCs” refers to the proportion of instances that end with an incomplete tool call containing an overly long plaintext PoC.
> - “grep/ls/find attempts” counts the agent actions used for simple retrieval operations (e.g., grep, ls, find). Failures correspond to those that produce empty output. The high failure rate supports our conclusion that the agent lacks more efficient retrieval tools.
>
> We will incorporate representative traces in the revision.
>
> For the third point regarding “flag guessing”, as noted in the paper (Line 932), CTF agent frameworks including Cybench and EnIGMA require flag feedback within their workflow. In our evaluation, a randomly generated “flag” is returned only if the candidate PoC passes the sanitizer-based oracle.

---

> > ### Comment · Reviewer_7oMB · 2025-11-24
> > **Official Comment from Reviewer 7oMB**
> >
> > Thank you for the additional information. I will increase my score.

---

> > > ### Author Response · Authors · 2025-11-25
> > >
> > > Thank you for raising the score. We appreciate your consideration and remain available for any further questions.

---

> > > > ### Author Response · Authors · 2025-11-28
> > > >
> > > > Thank you for your patience. We have uploaded a revised version of the paper, with all changes highlighted in blue. Below is a summary of our revisions, presented in a one-to-one correspondence with the points raised in our rebuttal above:
> > > >
> > > > - **Q1**: we added the additional qualitative and quantitative analysis of agent failure modes in Section 4 Line 461-464 and Appendix D Line 1360-1390, and corresponding representative example trajectories in Appendix F Figure 19,20,21.
> > > > - **Q2**: we incorporated the new analysis for data contamination including the statistical analysis and distribution comparison in Section 4 Line 374-387 and Appendix D Line 1204-1222.
> > > > - **Q3.1**: we added more detailed discussion about future work on expanding CyberGym’s scope in Section 6.
> > > > - **Q3.2**: we added the results and analysis of balanced resampling in Section 4 Line 342-344 and Appendix D Line 1223-1282.
> > > > - **Q4**: we added the human verification results and statistical analysis in Section 3.3 Line 268-272 and Appendix D Line 1039-1066.
> > > > - **Q5**: we explicitly mentioned the controlled iterations and budget in Section 4 Line 310-313, 325, 348-350, 359-361.
> > > > - **Q6**: we reported the success rates of different crash types in Appendix D Table 8.
> > > >
> > > > To ensure we have sufficient time to address any remaining concerns, we kindly ask whether our response has resolved the issues you raised. We remain fully available and would be glad to provide further clarification or make additional adjustments as needed.

---

### Official Review · Reviewer_Bi9L · 2025-10-28

**Soundness:** 4
**Presentation:** 4
**Contribution:** 4
**Rating:** 6
**Confidence:** 5

**Summary:**

The paper introduces CyberGym a large-scale benchmark designed to evaluate AI agents capabilities in real-world cybersecurity tasks. The core motivation is the inadequacy of current static, small-scale benchmarks, which fail to capture the dynamic, iterative nature of real-world security challenges. CyberGym addresses this by featuring 1,507 real-world vulnerabilities across 188 software projects, emphasizing a dynamic, interactive assessment where agents must iteratively generate and test exploits to achieve a successful outcome.

The authors demonstrate the utility of this benchmark by evaluating various current AI models(GPT, Claude, Gemini, DeepSeek and Qwen) on this challenging task set.

**Strengths:**

1. The use of 1,507 real-world vulnerabilities across 188 software projects is a substantial and necessary advancement over existing small-scale or synthetic benchmarks especially in the domain specific field of cubersecurity.

2. The requirement for dynamic and iterative problem-solving accurately reflects the complex and exploratory nature of real-world vulnerability research.

3. The intent to release a large, accessible dataset and environment is crucial for reproducibility in this field. The authors have done a good job committing to open-sourcing the dataset.

**Weaknesses:**

1. The authors must detail the computational cost required to run the full benchmark suite for future research teams.

2. The authors should have used a human cybersecurity analyst as a baseline reference to compare with the agents reference.

**Questions:**

n/a

---

> ### Author Response · Authors · 2025-11-21
>
> We sincerely thank the reviewer for their constructive feedback and will address each of their questions below.
> We are also updating the paper to incorporate all necessary revisions and will upload the revised version to OpenReview in a few days.
>
> **Q1: What is the computational cost required to run the full benchmark suite for future research teams?**
>
> The evaluation cost depends primarily on the agent framework and backbone model used. As detailed in Appendix C (Line 972), we control the average cost per instance to approximately \\$2 USD to ensure fair comparison across different agent frameworks with GPT-4.1. This results in a total evaluation cost of around \\$3,000 USD for the complete benchmark for one agent framework.
> To mitigate high costs, particularly when evaluating with expensive thinking mode models, we conducted some experiments on a randomly sampled 300-instance subset in the paper. We will publish the specific instance list to enable more lightweight evaluation on this reduced set.
>
> **Q2: How to provide a baseline reference to compare with the agents’ performance, such as human evaluation?**
>
> We will add a discussion on the human effort needed for vulnerability reproduction and related tasks. The task of vulnerability reproduction has been studied by [1]. Human security experts require about 5 hours to reproduce known vulnerabilities based on public reports, much longer when no usable PoC is available, as they have to manually analyze code and run various test cases. Prior research [2] has conducted empirical studies on the lifecycle of OSS-Fuzz-discovered vulnerabilities. Notably, the median time for fuzzing tools to reveal these vulnerabilities is 324 days, underscoring the challenge of detecting bugs in these real-world software projects. CyberGym serves as a critical foundation for developing and evaluating advanced cybersecurity agents to reduce human efforts in reproduction and enhance the security of software systems.
>
> [1] Mu, Dongliang, et al. "Understanding the reproducibility of crowd-reported security vulnerabilities." 27th USENIX Security Symposium (USENIX Security 18). 2018.
>
> [2] Keller, Brandon N., Benjamin S. Meyers, and Andrew Meneely. "What happens when we fuzz? Investigating OSS-fuzz bug history." 2023 IEEE/ACM 20th International Conference on Mining Software Repositories (MSR). IEEE, 2023.

---

> > ### Comment · Reviewer_Bi9L · 2025-11-24
> >
> > I thank the authors for their response and increase my score to 8.

---

> > > ### Author Response · Authors · 2025-11-25
> > >
> > > Thank you for raising the score. We appreciate your consideration and remain available for any further questions.

---

> > > > ### Author Response · Authors · 2025-11-28
> > > >
> > > > Thank you for your patience. We have uploaded a revised version of the paper, with all changes highlighted in blue. Below is a summary of our revisions, presented in a one-to-one correspondence with the points raised in our rebuttal above:
> > > >
> > > > - **Q1**: we provided the estimated cost and described the 300-sample subset used to mitigate the high overall cost in Section 4 Line 311-313.
> > > > - **Q2**: we discussed the difficulty and the human effort needed for vulnerability reproduction and related tasks in Section 3.2 Line 204-209.
> > > >
> > > > To ensure we have sufficient time to address any remaining concerns, we kindly ask whether our response has resolved the issues you raised. We remain fully available and would be glad to provide further clarification or make additional adjustments as needed.

---

### Official Review · Reviewer_yga9 · 2025-10-28

**Soundness:** 3
**Presentation:** 3
**Contribution:** 3
**Rating:** 8
**Confidence:** 4

**Summary:**

The paper proposes a benchmark (CyberGym) to evaluate the capabilities of LLM agents in reproducing exploits of previously known vulnerabilities. The authors curate 1507 vulnerabilities across 188 software projects sourced from Google’s OSS-Fuzz, and defined four tasks for each vulnerability with increasing level of difficulty based on the information availability. Interestingly, the paper shows that even though existing LLM agents struggle to accurately reproduce existing vulnerabilities (around 20%), they were still able to discover 35 zero-day vulnerabilities in the latest versions of the included projects.

**Strengths:**

- The thoroughness and rigor in collecting and filtering the dataset.

- The fact that the benchmark helped find 35 zero-day vulnerability is impressive.

- The paper includes useful and interesting sub-experiments and ablations, e.g., effect of thinking mode, and investigating data contamination.

**Weaknesses:**

- One minor weakness of the paper is that it does not sufficiently motivate the need for difficulty levels 1, 2, and 3. It is unclear to me when it useful to reproduce an exploit when we already have the stack trace from running the ground truth exploit (level 2), or when we have the ground truth patch (level 3). You mention that level 3 is useful in one-day settings, but to my knowledge, one-day settings are when the vulnerability is discovered but still not patched.

- The authors mention that considering the union of outcomes across several LLMs greatly improves the results. However, you did not show results for combining the best performing LLMs and agents (e..g., combining Claude-4, Claude-3.7, and GPT4, with thinking mode on). This could help show the performance upper bound of current systems.

**Questions:**

- What is the real-world benefit of benchmarking LLM agents when using difficulty levels 1, 2, and 3?

- Do your benchmark include a way to evaluate whether a patch is successful by ensuring that the original program functionality is maintained? If no, do you think adding such feature to benchmarks is challenging?

- How accessible is your benchmark? For instance, how much would it cost (dollars, or GPU hours) to evaluate a new LLM or agentic approach? Are there ways to mitigate the potentially high evaluation cost (e.g., provide a smaller subset of the benchmark)?

- You mention in Appendix E a case where one vulnerability was patched over several commits. Is this a failure of your benchmark, where you should have detected that and included this in the ground truth information of that vulnerability? Does this affect the accuracy of your benchmark?

- How do you plan to deal with the mentioned cases when the LLM defers actions and asks for user confirmation? This could mistakenly skew the results.

- In total, how many of the 1507 vulnerabilities were reproduced by at least one LLM? In other words, do all LLMs detect the same vulnerabilities, or does combining their outcomes bring more gains? (check weakness 2)

- Did you test running level 0 tasks (reproducing the vulnerability with no additional information) on known existing vulnerabilities? This could simulate the process of rediscovering zero-day vulnerabilities in historical data, especially when evaluating on vulnerabilities found after the knowledge cutoff of the LLM.

- CyberGym was able to discover zero-day vulnerabilities using the 759 PoCs/exploits obtained by reproducing existing vulnerabilities. I am confused as to how the resulting vulnerabilities are considered zero-day when their ground PoCs already exist?


Minor issues:

- In lines 426 and 437, the referred results are in Appendix E not D.

---

> ### Author Response · Authors · 2025-11-21
>
> We sincerely thank the reviewer for their constructive feedback and will address each of their questions below.
> We are also updating the paper to incorporate all necessary revisions and will upload the revised version to OpenReview in a few days.
>
> **Q1: What is the real-world benefit of benchmarking LLM agents when using difficulty levels 1, 2, and 3? Could you clarify the one-day setting for level 3?**
>
> We clarify the practical motivation for each difficulty level and address the confusion about 1-day vulnerabilities below:
> - Level 1 (textual description only): Many community vulnerability reports such as CVEs only provide textual vulnerability descriptions without working PoC [1]. Security researchers must reconstruct PoCs from these descriptions, which costs significant effort. Our level 1 evaluates whether AI agents can bridge the gap between textual vulnerability reports and working PoCs.
> - Level 2 (+ crash stack traces): As noted in [1]’s survey, *“18 respondents believed that information about the exact location of the vulnerable code was necessary for a report to be complete.”* The crash stack traces emulate the scenario by supplying additional context about the exact location of the vulnerabilities. This level evaluates whether agents can leverage these supplementary location details to more effectively construct working PoCs.
> - Level 3 (+ patch information, 1-day setting): While the definition of "1-day" is not so uniform as "0-day", prior work  [2,3,4,5] typically uses "1-day" to refer to disclosed vulnerabilities for which patches exist but have not yet been widely deployed. When vendors release security patches, the patches themselves reveal vulnerability details through code diffs. Attackers routinely perform "patch analysis" to reverse-engineer vulnerabilities and develop PoCs targeting unpatched systems, which has been actively studied for decades. This level tests whether agents can automate this patch-to-exploit process, which is critical for both red teams (testing defenses) and blue teams (understanding their exposure before patches are deployed).
>
> These three levels capture the spectrum of information availability in real-world vulnerability disclosure and response. They allow us to measure how information richness affects agent performance and identify which scenarios pose the greatest automation risks or opportunities for defensive applications. We will revise the paper to discuss this.
>
> [1] Mu, Dongliang, et al. "Understanding the reproducibility of crowd-reported security vulnerabilities." 27th USENIX Security Symposium (USENIX Security 18). 2018.
>
> [2] Oh, Jeongwook. "Fight against 1-day exploits: Diffing binaries vs anti-diffing binaries." Blackhat technical security conference. 2009.
>
> [3] Duan, Ruian, et al. "Identifying open-source license violation and 1-day security risk at large scale." Proceedings of the 2017 ACM SIGSAC Conference on computer and communications security. 2017.
>
> [4] Woo, Seunghoon, et al. "{V1SCAN}: Discovering 1-day Vulnerabilities in Reused {C/C++} Open-source Software Components Using Code Classification Techniques." 32nd USENIX Security Symposium (USENIX Security 23). 2023.
>
> [5] Yang, Songtao, et al. "1dfuzz: Reproduce 1-day vulnerabilities with directed differential fuzzing." Proceedings of the 32nd ACM SIGSOFT International Symposium on Software Testing and Analysis. 2023.
>
>
> **Q2: Does the benchmark include a way to evaluate whether patches maintain original program functionality, and if not, how challenging would this be to add?**
>
> No. CyberGym does not currently include a mechanism to evaluate whether patches preserve original program functionality. Adding such a capability is valuable but requires significant effort, and we consider it as a future work item:
>
> Projects in the benchmark rely on heterogeneous test systems (e.g., CMake, Makefiles, ad-hoc scripts, manually written test drivers). Prior works have had to extract and standardize these test setups manually, which is labor-intensive and error-prone [1,2].
> Reliably generating new, high-quality test cases that can assess functional correctness remains an open research problem. Although widely studied, automated test generation, especially for complex, real-world projects, still faces substantial challenges in coverage, oracle quality, and robustness [3].
>
> [1] Dilgren, Connor, et al. "Secrepobench: Benchmarking llms for secure code generation in real-world repositories." arXiv preprint arXiv:2504.21205 (2025).
>
> [2] Chen, Junkai, et al. "SecureAgentBench: Benchmarking Secure Code Generation under Realistic Vulnerability Scenarios." arXiv preprint arXiv:2509.22097 (2025).
>
> [3] Wang, Junjie, et al. "Software testing with large language models: Survey, landscape, and vision." IEEE Transactions on Software Engineering 50.4 (2024): 911-936.

---

> ### Author Response · Authors · 2025-11-21
>
> **Q3: How accessible is the benchmark in terms of evaluation cost (dollars or GPU hours), and are there ways to mitigate high costs such as providing a smaller subset?**
>
> The evaluation cost depends primarily on the agent framework and backbone model used. As detailed in Appendix C (Line 972), we control the average cost per instance to approximately \\$2 USD to ensure fair comparison across different agent frameworks with GPT-4.1. This results in a total evaluation cost of around \\$3,000 USD for the complete benchmark for one agent framework.
> To mitigate high costs, particularly when evaluating with expensive thinking mode models, we conducted some experiments on a randomly sampled 300-instance subset in the paper. We will publish the specific instance list to enable more lightweight evaluation on this reduced set.
>
> **Q4: How do you treat cases where one vulnerability was patched over several commits, such as the one in Appendix E?**
>
> During our experiments, we found 17 cases with incomplete patches, representing only ~1% of our 1,507 benchmark instances. We have corrected these cases by replacing the post-patch version with the version that finally addressed the target vulnerability. We believe this strengthens the quality of our benchmark.
>
> **Q5: How to deal with cases when the LLM defers actions and asks for user confirmation, which could skew results such as for o4-mini?**
>
> To ensure a fair comparison across all models, we report results using the same prompt configuration and evaluation setup for every model in our experiments.
> For o4-mini, we investigated its behavior. When it asks for user confirmation, it often terminates the agent execution by calling the [`finish` tool](https://github.com/OpenHands/OpenHands/blob/1a3360698779ec993cb8ae76b7e56fdbf56f4939/openhands/agenthub/codeact_agent/tools/finish.py) in OpenHands. This breaks OpenHands’s mechanism to insert a dummy [“user approval” message](https://github.com/OpenHands/OpenHands/blob/1a3360698779ec993cb8ae76b7e56fdbf56f4939/openhands/core/main.py#L314-L327) to continue the execution. We attempted minor prompt adjustments to force o4-mini to proceed without asking for confirmation, but these did not reliably eliminate the behavior.
> We did not observe this pattern in other evaluated models. Thus, this appears to be a model-specific limitation of o4-mini in this scenario.
> We believe that when developing agents for real-world deployment, it is the responsibility of developers to address such model-specific behaviors as part of robust agent design.
>
> **Q6: How many of the 1507 vulnerabilities were reproduced by at least one LLM, and what are the results for combining the best performing LLMs to show the performance upper bound?**
>
> To understand the performance upper bound, we analyzed the union of successful reproductions across different configurations:
> - all models in Figure 3: 410 (27.2%)
> - all agent frameworks in Figure 5: 277 (18.4%)
> - all models and agent frameworks: 456 (30.3%)
> - all models, agent frameworks, and models with extended thinking: 490 (32.5%)
>
> **Q7: Did you test running level 0 tasks on known existing vulnerabilities to simulate rediscovering zero-day vulnerabilities, especially for those found after the LLM's knowledge cutoff?**
>
> Yes, we evaluated level 0 tasks on known existing vulnerabilities using OpenHands with GPT-4.1 to assess the agent's ability to rediscover real vulnerabilities.
> The agent achieved 77/1507 (5.11%) successful crashes overall, with 69/1365 (5.05%) on vulnerabilities before the knowledge cutoff and 8/142 (5.63%) on those discovered after the cutoff.
>
> **Q8: How are the vulnerabilities discovered using the 759 PoCs/exploits considered zero-day when their ground truth PoCs already exist?**
>
> As mentioned in Line 431, these 759 PoCs, which are newly generated during our experimentation, still cause a crash on the *post-patch* version of the program. This indicates that they do not reproduce the intended vulnerability as the ground truth PoCs, because successful reproduction requires not crashing the post-patch version. Instead, they expose a different vulnerability in the post-patch version. To further determine whether these represent zero-day vulnerabilities, we check (i) whether the crashes also occur in the latest program version, and (ii) whether the issues have already been publicly reported. After 10 crashes were confirmed as unique zero-days.

---

> > ### Author Response · Authors · 2025-11-28
> >
> > Thank you for your patience. We have uploaded a revised version of the paper, with all changes highlighted in blue. Below is a summary of our revisions, presented in a one-to-one correspondence with the points raised in our rebuttal above:
> >
> > - **Q1**: we added explanations about the real-world benefit of different difficulty levels in Section 3.2.
> > - **Q2**: we discussed the challenge of how to evaluate the patch as future work in Section 6.
> > - **Q3**: we provided the estimated cost and described the 300-sample subset used to mitigate the high overall cost in Section 4 Line 311-313.
> > - **Q4**: we explicitly mentioned we addressed these incomplete patches in our benchmark in Section 5 Line 497.
> > - **Q5**: we added discussion about the behavior of requesting user confirmation for different models in Section 4 Line 332-339.
> > - **Q6**: we added the union results in Section 4 Line 340-343.
> > - **Q7**: we added these results to the paragraph comparison of different difficulty levels in Section 4 Line 397-402.
> > - **Q8**: we added clarifications that the generated PoCs which crash the post-patch versions differ from the ground truth PoCs in Section 5 Line 481-483.
> >
> >
> > To ensure we have sufficient time to address any remaining concerns, we kindly ask whether our response has resolved the issues you raised. We remain fully available and would be glad to provide further clarification or make additional adjustments as needed.
> >
> > Thank you very much for your time and consideration, and we look forward to your response.

---

### Official Review · Reviewer_cQYX · 2025-10-30

**Soundness:** 4
**Presentation:** 4
**Contribution:** 4
**Rating:** 8
**Confidence:** 4

**Summary:**

CyberGym is a benchmark for agents to discover vulnerabilities in OSS projects.
- Sourced 1507 vulneabilities from OSS-Fuzz (spanning 2017-2025, a chunk beyond most frontier models' cutoff - data contamination assessment shows limited contamination)
- Contains containerized pre-patch codebase, post-patch codebase, ground truth PoC (produced by OSS-Fuzz), and ground truth patch
- Agent must produce a Proof-of-concept bash script that (1) Triggers sanitizer crash in pre-patched version (2) Do not trigger sanitizer crashes in post-patched versions.
- The codebases have median of 1k files, and ground truth PoCs

Levels 0-3 of information provided to the agent:
- Prepatch codebase only (open-ended) < +Vul. Text description < +Ground truth PoC and Stack trace < Ground truth patch (diff)

Experiments with OpenHands agent scaffolding.

Some interesting results:
- Specialized models with strong results on SWE bench may actually perform poorly on Cyberbench
- Best results with Sonnet-4 and GPT-4.1
-

**Strengths:**

The authors seem to have taken strong steps towards reproducibility by providing the pre-patched program in a containerized environment.
The vulnerabilities are present in real open source projects, which makes it a very realistic benchmark.
The authors discover zero-day vulnerabilities in the Level 0 mode of the benchmark; this means the approach taken in the benchmark can be used by maintainers to discover new vulnerabilities.

**Weaknesses:**

Most of the vulnerabilities seem to be related to C/C++ bad memory usage. While some of the projects are extremely popular, it should be made clear that it is really a subset of cyber-risks. "Cybersecurity capabilities" might be over-selling it a bit.

Success of PoC is taken as crashing the sanitizer. I understand why the authors made this choice, but again, this is a narrow subset of vulnerabilities and I think the paper can be more explicit regarding the limitations of scope.

**Questions:**

Which level is Figure 3? I assume it's level 1 from cross-referencing w/ figure 6 but it should be explicit.

---

> ### Author Response · Authors · 2025-11-21
>
> We sincerely thank the reviewer for their constructive feedback and will address each of their questions below.
> We are also updating the paper to incorporate all necessary revisions and will upload the revised version to OpenReview in a few days.
>
> **W1 and W2: Given that the benchmark focuses on C/C++ memory safety issues with sanitizer crashes, can the authors better acknowledge the resulting limitations in scope?**
>
> Yes, we will revise our paper accordingly as suggested. In particular, we will clarify that our focus only covers a slice of the broader cybersecurity landscape. We will also add an explanation on why we selected this scope.
>
> **Q1: Which difficulty level is evaluated in Figure 3?**
>
> We evaluate level 1 difficulty (as specified in Line 284), our primary reproduction task, in Figure 3,4,5,7. And we compare the performance across different difficulty levels in Figure 6.
> We will explicitly annotate the evaluated difficulty level in each figure to further enhance clarity.

---

> > ### Author Response · Authors · 2025-11-28
> >
> > Thank you for your patience. We have uploaded a revised version of the paper, with all changes highlighted in blue. Below is a summary of our revisions, presented in a one-to-one correspondence with the points raised in our rebuttal above:
> > - **W1 and W2**: we added a paragraph in Section 3.1 discussing our chosen scope, explaining the rationale behind this focus and acknowledging its limitations.
> > - **Q1**: we explicitly highlighted the corresponding difficulty levels in Figure 3,4,5,7
> >
> > To ensure we have sufficient time to address any remaining concerns, we kindly ask whether our response has resolved the issues you raised. We remain fully available and would be glad to provide further clarification or make additional adjustments as needed.
> >
> > Thank you very much for your time and consideration, and we look forward to your response.

---

### Author Response · Authors · 2025-12-01

Dear Chairs,

We acknowledge the decision to revert review scores and close the author-reviewer discussion due to the OpenReview security incident. We sincerely appreciate your efforts in managing ICLR during this challenging situation.

We would like to provide a brief summary of the progress made during the rebuttal phase of our paper prior to the incident becoming publicly known on November 28. Before November 25, Reviewers 7oMB and Bi9L explicitly acknowledged our rebuttal and increased their scores from 6 to 8. Although Reviewers cQYX and yga9 did not have the opportunity to respond before the discussion closed, we addressed all of their questions. We have incorporated all review feedback into the latest revised version of our paper on OpenReview.

Best regards,

Authors

---

### Meta-Review · Area_Chair_MQdS · 2025-12-21

**Summary:**

The paper proposes CyberGym, a large-scale, execution-based benchmark for evaluating the cybersecurity capabilities of AI agents. The reviewers were unanimous in their appreciation for the benchmark's scale (1,507 vulnerabilities), its realism compared to static Q&A datasets, and its demonstrated utility in discovering zero-day vulnerabilities. Initial concerns primarily revolved around the benchmark's heavy focus on C/C++ memory safety issues (potentially limiting its scope), the high computational cost of running the full evaluation, and the rigor of the data contamination analysis. The authors provided an effective rebuttal, offering cost estimates and a representative subset for reproducibility, clarifying the scope, and providing robust statistical evidence regarding data contamination.

**Reviewer Concerns:**

### Addressed:
- Data Contamination (Reviewer 7oMB): The reviewer rigorously challenged the data contamination analysis. The authors successfully addressed this by providing Fisher’s exact tests and z-tests showing no significant performance difference pre/post knowledge cutoff.

- Agent Failure Modes (Reviewer 7oMB): The authors added a qualitative analysis of agent behaviors (e.g., struggling with basic retrieval tools), which added necessary depth to the paper.

- Human Baselines (Reviewer Bi9L): The authors added context regarding human effort to better frame the agent performance.

### Outstanding:

- Scope vs. Branding (Reviewers cQYX, yga9, 7oMB): While the authors updated the text to acknowledge the focus on C/C++ memory safety, the benchmark itself remains functionally limited to this domain. The name "CyberGym" implies a broader coverage (e.g., web, network, logic bugs) that is not present.

**Reviewer Scores:**

- Reviewer cQYX: 8

- Reviewer yga9: 8

- Reviewer Bi9L: 6 -> 8

- Reviewer 7oMB: 6 -> 8

---

### Decision · Program_Chairs · 2026-01-26

Accept (Oral)